# Epidemiology and Trends of Cutaneous Fungal Infections (2019–2022) in Israel: A Single Tertiary-Center Study

**DOI:** 10.3390/jof11040320

**Published:** 2025-04-18

**Authors:** Eran Galili, Auriella Taieb, Avner Shemer, Gil Leor, Anna Lyakhovitsky, Aviv Barzilai, Sharon Baum

**Affiliations:** 1Department of Dermatology, Sheba Medical Center, Ramat-Gan 52621, Israel; auriellat@mail.tau.ac.il (A.T.); ashemer1@gmail.com (A.S.); annalyderm@gmail.com (A.L.); aviv.barzilai@sheba.health.gov.il (A.B.); sharon.baum@sheba.health.gov.il (S.B.); 2Faculty of Medicine and Health Sciences, Tel Aviv University, Tel Aviv 69978, Israel; gilleor23@gmail.com

**Keywords:** cutaneous fungal infection, dermatophytes, *Trichophyton rubrum*, *Trichophyton tonsurans*

## Abstract

Cutaneous fungal infections predominantly caused by dermatophytes are a global concern. These infections vary widely by region, age, and body site, with recent shifts in the pathogen distribution. This study examines the distribution and trends of superficial fungal infections in a large tertiary care center in Israel from 2019 to 2022. A retrospective analysis of 2244 patients with suspected fungal infections was performed, utilizing PCR and fungal cultures for diagnosis. Confirmed fungal infections were present in 53.0% of cases. In adults, infections predominantly affected the nails and feet, while in children, the scalp and nails were the most involved sites. *Trichophyton rubrum* was the most common pathogen overall, but *T. tonsurans* was the leading cause of scalp, face, and neck infections, as well as tinea corporis in children. *T. tonsurans* incidence significantly increased in adults and became the most frequent agent of upper-body tinea corporis by 2022. These findings highlight a shift in pathogen distribution, with *T. tonsurans* emerging as the leading cause of upper-body skin infections, underscoring the need for targeted prevention strategies and further investigation of its transmission routes.

## 1. Introduction

Superficial cutaneous fungal infections are among the most common dermatological diseases worldwide [1]. The causative pathogens are mainly dermatophytes, but also yeasts (e.g., *Candida* spp.) and, to a lesser extent, non-dermatophyte molds (*NDMs*) such as *Scopulariopsis*, *Fusarium*, and *Aspergillus* spp. Dermatophytes belong predominantly to three genera: *Trichophyton*, *Microsporum*, and *Epidermophyton* [2]. The global prevalence rate is estimated to be between 20 and 25% [3]. In immunocompetent individuals, cutaneous fungal pathogens can invade keratinized tissues, the epidermis, appendages, nails, and hair, causing diverse clinical manifestations. While cutaneous fungal infections in immunocompetent individuals typically remain superficial, immunocompromised patients, such as those with HIV, malignancies, or undergoing immunosuppressive therapy, may develop more extensive or atypical presentations [4,5]. In rare cases, deep dermatophytosis or disseminated fungal infection can occur, particularly in patients with primary immunodeficiencies, such as CARD9 (Caspase Recruitment Domain-containing protein 9) deficiency [5,6].

Cutaneous fungal infections exhibit diverse epidemiology across different regions and are influenced by various factors, including age, sex, geographical location, and socioeconomic status [1,3,7]. Superficial fungal infections are more prevalent in males than in females, particularly in children [1]. The location of infection is also influenced by age, with tinea capitis being the predominant fungal infection among children [8] and onychomycosis predominating in adults [7]. Understanding the distribution and trends of these infections is important for effective disease management. In recent decades, temporal shifts in the distribution of cutaneous fungal pathogens have been reported [2,9]. One notable emerging trend is the rise in *Trichophyton tonsurans*, an anthropophilic dermatophyte species that is mainly known for causing endothrix tinea capitis but is also capable of infecting other skin sites. *T. tonsurans* is the leading pathogen causing tinea capitis among children and adults in many countries, including the United States and United Kingdom [10,11]. In Israel, a 15-year retrospective study by Friedland et al. reported *T. tonsurans* as the predominant cause of pediatric tinea capitis by 2020 [12], while Shemer et al. similarly observed its predominance among pediatric patients with tinea capitis during the 2010s [13]. However, less is known about the epidemiology of *T. tonsurans* in non-scalp sites. A recent study from the United States identified *T. tonsurans* as a leading cause of tinea corporis, whereas other reports suggest its presence in other anatomical sites, such as the face and nails, is uncommon [14,15,16,17,18]. Trends in pathogen distribution are also influenced by surges in treatment-resistant fungi, such as *T. indotineae*. This newly defined pathogen belonging to the *T. mentagrophytes*/*T. interdigitale* complex has evolved to be the most common cause of tinea corporis in India and is also rapidly expanding worldwide [19,20].

Accurate diagnosis of cutaneous fungal infections is required for appropriate treatment. Laboratory methods, such as microscopy, allow fungal detection, whereas other methods, such as fungal culture, and molecular techniques, such as Polymerase Chain Reaction (PCR), are used to identify the causative fungal species [3,21]. Previous studies have demonstrated that PCR-based methods offer superior sensitivity and specificity, as well as faster species identification, compared to fungal culture in skin and nail specimens [21,22]. However, commercial PCR-based kits, which are increasingly being adopted in clinical laboratories, have several disadvantages compared to fungal culture, including higher cost, inability to assess fungal viability, and detection limited to a predefined range of fungal species [23].

In this study, we aimed to delineate the epidemiology of cutaneous fungal infections among children and adults in a large tertiary care center. Additionally, we aimed to identify the shifts in pathogen distribution in recent years.

## 2. Materials and Methods

This retrospective study included patients with cutaneous lesions suspected of fungal infections at the Dermatology Department of Sheba Medical Center, Israel, between 2019 and 2022. Sheba Medical Center, located in central Israel, is the largest hospital in the country and serves as a national tertiary referral center. The majority of patients are referred from the central region of Israel.

Demographic information, including age, sex, and the anatomical site of the skin specimen, was retrieved from the patients’ medical files. Skin and hair specimens were collected from the suspected lesions, and fungal identification was performed in two consecutive steps. At first, a Polymerase Chain Reaction (PCR) commercial kit for dermatophytes was applied (DERMADYN multiplex PCR kit, Dyn Diagnostics, Caesarea, Israel [24]). This kit was designed to identify seven dermatophyte species: *Epidermophyton fluccosum*, *Microsporum canis*, *trichophyton mentagrophytes*\*interdigitale complex*, *T. rubrum*, *T. tonsurans* and *T. violaceum*. Specimens that tested negative using the PCR kit were reassessed through direct microscopy and fungal culture, as deemed by the treating dermatologist. This approach allowed the detection of yeasts, non-dermatophyte molds, and dermatophytes that were not covered in the PCR kit. Confirmed cutaneous mycosis was defined as identification of a fungal pathogen via PCR and/or fungal culture.

The rationale behind this stepwise approach was to harness the superior speed and detection accuracy of PCR for the commonly encountered dermatophytes covered by the kit, while using microscopy and culture in PCR-negative cases to identify uncommon species and non-dermatophytes. The rationale for this stepwise approach was to utilize PCR first, given its superior speed and accuracy in detecting common dermatophyte species included in the kit compared to fungal culture. For cases where PCR results were negative, microscopy and fungal culture were used to detect uncommon species and non-dermatophyte fungi, which the PCR kit may miss due to its limited coverage of species.

A dermatologist selected the sampling site; in some cases, multiple anatomical sites were sampled. Positive specimens were clinically categorized based on the anatomical site from which they were obtained. Data on negative specimens were also collected. Ethical approval was obtained from the Research Ethics Committee of Sheba Medical Center (no. SMC-7082-20).

### Statistical Analysis

Values are presented as mean (±standard deviation) or number (percentage). Categorical variables were compared using the chi-square test. Agreement between PCR, culture, and microscopy was assessed using the kappa index, interpreted as follows: <0.20 = slight, 0.21 − 0.40 = fair, 0.41 − 0.60 = moderate, 0.61 − 0.80 = substantial, and >0.80 = almost perfect agreement. All statistical tests were two-sided, and *p*-values < 0.05 were considered statistically significant. Statistical analyses and graphs were generated using R, version 4.2.2.

## 3. Results

### 3.1. Baseline Characteristics and Clinical Features

In total, 2244 patients presenting with cutaneous lesions suggestive of fungal infection were assessed in this study. Overall, 2800 dermatophyte PCR kits, 905 fungal cultures, and 897 microscopic tests were conducted. A total of 1190 patients with 1542 infected skin sites positive for fungal pathogens isolated by PCR or fungal culture were recruited for this study. The detection rates of fungal pathogens by PCR and culture were 50.3% and 49.7%, respectively. Microscopic results were positive in only 24.2% of tested cases.

### 3.2. Age and Sex Distribution

The demographic profile, reported in Table 1, revealed that patients diagnosed with dermatomycosis had a mean age of 44.3 years (±21.6). Patients detected positive for fungal infection were mainly adults older than 18 years old (94.9%, *n* = 1134; mean age 46.1 [±20.7] years), with a minority (5.1%, *n* = 56; mean age 11.0 [±4.7] years) being children. In terms of sex distribution, 61.0% and 69.6% of the adult patients and children were male, respectively. Table 1 presents an overview of the age and sex distributions across different sites of infection. Notably, the oldest age group belonged to patients with isolates from crural area (encompassing lesions on the buttocks, groin, inner thighs, and genitals; mean age 49.8 [±22.0] years) and trunk and limbs (mean age of 48.3 [±24.0] years), while the youngest age group belonged to patients with scalp (mean age of 19.8 [±13.3] years) and face and neck (mean age of 32.4 [±18.7] years) isolates. Males outnumbered females at all isolated sites, with the largest sex differences observed in the scalp and palmar isolates (85.9% and 79.2% males, respectively).

### 3.3. Infected Body Site and Positive Detection Rate

In adults, positive specimens were most commonly isolated from the nails (44.1%), followed by the feet (38.7%) and the trunk and limbs (11.9%). In children, the most common sites of isolation were the scalp (32.9%), nails (27.1%) and feet (14.3%). The likelihood of a positive result in specimens varied by anatomical site, with the highest rates observed in tinea pedis (70.8%) and onychomycosis (61.4%), and lower rates in the palm (34.3%) and the trunk and limbs (33.8%), as detailed in Table 1.

Among the 808 samples tested by both PCR and fungal culture, a subset of 745 samples either tested negative or included dermatophyte species targeted by the PCR kit. Within this group, 530 samples (71.1%) demonstrated concordant results between the two methods. Among PCR-negative specimens, 28.3% were subsequently found to be positive by fungal culture. The calculated Cohen’s kappa index was 0.41, indicating moderate agreement between PCR and culture for the detection of dermatophyte species included in the PCR kit. Comparison between PCR and microscopy, based on 741 cases, yielded a Cohen’s kappa of 0.54, while microscopy versus fungal culture, assessed in 823 cases, showed a kappa of 0.48, both reflecting moderate levels of concordance.

### 3.4. Epidemiology of Fungal Pathogens by Site and Age Group

Table 2 and Table 3 show fungal species distribution by body site and age group. *T. rubrum* was the most frequently isolated pathogen in adults and children (80.4% and 51.4%, respectively), followed by *T. tonsurans* (9.2% and 28.6%, respectively). Other fungal species have rarely been reported in adults, accounting for less than 2.5% of the isolates. In contrast, among children, *M. canis* was also common (15.7%), whereas other fungal species were infrequently reported, each contributing less than 1.5% of the isolates.

Among adults, *T. rubrum* was the most common causative pathogen of onychomycosis, accounting for 87.4% of the cases (*n* = 570), followed by *candida* spp. (4.8% [*n* = 30], predominantly *C. parapsilosis* [*n* = 28], in 93.3% of the cases). Other pathogens included NDM (2.5%, *n* = 16), *E. floccosum* (2.5%, *n* = 16), and *t. mentagrophytes*/*interdigitale* complex (1.5%, *n* = 10). *T. rubrum* was also the most common pathogen causing tinea pedis (92.2%, *n* = 357), followed by *E. floccusom* (3.6%, *n* = 14) and *NDM* (1.6%, *n* = 6). It was also the leading cause of crural infections (80.4%, *n* = 90), with *T. tonsurans* (5.4%, *n* = 60) and *C. albicans* (5.4%, *n* = 6) being the next most common pathogens. For tinea manuum, *T. rubrum* accounted for 65.2% (*n* = 30) of cases, followed by *T. tonsurans* and *C. parapsilosis*, each contributing 10.9% (*n* = 5). Similarly, it was the most frequent cause of tinea corporis (67.6%, *n* = 119) with *T. tonsurans* (25.6%, *n* = 45) and *M. canis* (2.8%, *n* = 5) being the next most common pathogens. *T. tonsurans* was the most common pathogen causing tinea capitis, accounting for 83.3% (*n* = 40) of the cases, followed by *T. rubrum* (8.3%, *n* = 4) and *T. violaceum* (6.3%, *n* = 3). It was also the leading cause of head and neck infections, accounting for 66.7% (*n* = 35) of the cases, with *T. rubrum* (25.5%, *n* = 13) and *M. canis* (3.9%, *n* = 2) being the second and third most common pathogens, respectively.

Among children, *T. rubrum* was the sole causative pathogen identified in cases of onychomycosis, tinea pedis, and tinea manuum (*n* = 19, 10, and 2, respectively). In contrast, *T. tonsurans* was the most common pathogen causing tinea corporis (71.4%, *n* = 5), followed by *T. rubrum* and *M. canis*, which accounted for 14.3% (*n* = 1) of the cases. *T. tonsurans* was also the leading cause of tinea capitis (47.8%, *n* = 11), with *M. canis* (39.1%, *n* = 9) and *T. rubrum* (8.7%, *n* = 2) as the second and third most common pathogens, respectively. Similarly, *T. tonsurans* was the predominant pathogen in head and neck infections (57.1%, *n* = 4), followed by *T. rubrum* (28.6%, *n* = 2), and *M. canis* (14.3%, *n* = 1).

### 3.5. Trends over the Study Period

During the study period, from 2019 to 2022, the annual incidence of *T. tonsurans* increased significantly among adults, rising from 4.8% (*n* = 22/454) in 2019 to 11.5% (*n* = 45/390) in 2022 (*p* < 0.001), as shown in Figure 1. Although a general increase in *T. tonsurans* infections was observed from 2019 to 2022, a slight decrease occurred in 2022 compared to 2021 (from 12.7% to 11.5%), which was not statistically significant (*p* =0.64). Among children, the incidence increased non-significantly from 30.8% (*n* = 4/13) in 2019 to 40.3% (*n* = 10/23) in 2022 (*p* =0.5), as shown in Figure 1. When trends were analyzed based on the site of infection (Figure 2), a consistently high percentage of *T. tonsurans* isolates was observed in the upper-body regions, including the scalp, head and neck, trunk, and upper limbs, compared to a low percentage of isolates from the lower body regions, including the crural region, legs, feet, and nails. A statistically significant increase in head and neck *T. tonsurans* infections among adults was observed, from 30.0% in 2019 (*n* = 3/10) to 78.6% in 2022 (*n* = 10/14) (*p* < 0.05). Additionally, among adults, there was a non-significant increase in *T. tonsurans* upper-body tinea corporis (involving the trunk and upper limbs, excluding the palms) from 20.0% in 2019 (*n* = 4/20) to 46.4% in 2022 (*n* = 16/28) (*p* = 0.11), making it the leading causative pathogen in 2022. Interestingly, over the entire study period, *T. tonsurans* was a significantly more frequent cause of upper-body tinea corporis than lower-body tinea corporis (38.9% [*n* = 42/108] vs. 10.7% [*n* = 8/75], respectively; *p*-value < 0.001).

## 4. Discussion

This study retrospectively evaluated the epidemiology of cutaneous fungal infections in a large tertiary medical center in Israel over a four-year period. The findings highlight critical trends in pathogen distribution. *T. rubrum* remained the predominant species, consistent with global patterns, particularly in cases of onychomycosis, tinea pedis, and tinea manuum [1]. Its continued dominance reinforces its well-established role as the leading dermatophyte worldwide [1]. In contrast, a marked rise in *T. tonsurans* among both children and adults was observed during the study period. While *T. tonsurans* is well recognized as a primary cause of tinea capitis, as previously reported in Israel and other regions [13,25,26], our findings suggest its increasing importance as a leading cause of dermatophytoses affecting the head, neck, and upper body. Throughout the study period, *T. tonsurans* was the most common cause of face and neck infections, as well as the leading and second most common cause of upper-body tinea corporis in children and adults, respectively. By 2022, *T. tonsurans* had become the leading cause of upper-body tinea corporis in adults.

To our knowledge, no previous study has reported an increasing trend in upper-body tinea corporis and tinea faciei attributable to *T. tonsurans*. Supporting our findings, a recent analysis from a major commercial laboratory in the United States identified *T. tonsurans* as the predominant etiologic agent of tinea corporis [18]. Similarly, a small-scale study conducted in Munich, Germany, analyzed 447 dermatophyte-positive skin samples, including 37 positive for *T. tonsurans*, collected during a similar time period (2019–2022) [27]. That study reported a 10-fold increase in the overall proportion of *T. tonsurans*, along with a 2-fold increase in *T. tonsurans* tinea capitis and a 4-fold increase in *T. tonsurans* tinea corporis [27]. However, neither study stratified tinea corporis cases by anatomical region, nor did they assess tinea faciei. In contrast, several recent studies investigating the epidemiology of tinea faciei did not observe a comparable trend [14,15,16,17]. However, these studies were conducted in regions where *T. tonsurans* is not considered endemic, potentially explaining the discrepancy [14,15,16,17]. Focusing on children, our study found that *T. tonsurans* was frequently isolated from non-scalp sites, including 57.1% of face and neck cases and 71.4% of trunk infections, indicating broader clinical involvement than traditionally associated with scalp disease [8,28,29]. Only a few studies have reported *T. tonsurans* infections in non-scalp sites among children, and none have been epidemiological in nature [30,31,32]. Although a general increase in *T. tonsurans* infections was observed from 2019 to 2022, a slight decrease occurred in 2022 compared to 2021, which was not statistically significant. This may reflect normal annual variability, and ongoing surveillance is required to determine future trends.

While *T. tonsurans* tinea capitis was previously reported to be caused mostly by infected hairdresser utensils and practicing contact sports, *T. tonsurans* infection of the face, neck, upper limbs, and trunk observed in our study may reflect broader transmission routes [10,33,34]. This involvement could be a consequence of inter- and intrapersonal contact transmission between symptomatic and asymptomatic carriers of scalp *T. tonsurans* [35,36,37]. Supporting this, *T. tonsurans* was notably absent in infections such as onychomycosis, tinea pedis, and lower-limb lesions. The distinct anatomical distribution of *T. tonsurans* indicates a possible reservoir on the scalp, reinforcing the need for targeted hygiene practices and prevention strategies. Previous studies have reported tinea corporis as common among young adults with *T. tonsurans* tinea capitis [13], suggesting that untreated or inadequately treated scalp infections facilitate the spread of fungi to other body sites. Further research is needed to determine whether an antifungal shampoo is indicated in patients with upper-body *T. tonsurans* lesions by addressing a potential scalp *T. tonsurans* reservoir.

Sex differences observed in this study are consistent with existing literature [1]. Males were disproportionately affected by tinea capitis and tinea cruris. Hormonal, occupational, and behavioral factors may contribute to this disparity.

This study also revealed that the clinical suspicion of fungal infections by referring dermatologists was frequently inaccurate, except for the nails, feet, and buttocks. Specimens from other anatomical sites such as the palms, face, neck, trunk and limbs mostly yielded negative results. These findings highlight the importance of judicious use of antifungal therapies, particularly at certain anatomical sites, to avoid inadequate treatment and treatment resistance. Notably, although PCR has previously been reported to offer superior detection rates, 28% of the PCR-negative specimens that were subsequently evaluated by fungal culture were found to be positive for dermatophyte species that are included in the PCR kit [21,23]. In our study, the concordance between PCR and fungal culture was moderate, with a Cohen’s kappa index of 0.41. This level of agreement indicates that while both methods are valuable, they are not interchangeable. These findings underscore the importance of a combined diagnostic approach, particularly in cases with high clinical suspicion, to ensure accurate pathogen detection. Only 2.8% of PCR-negative cases and 1.8% of culture-negative cases were found to be positive by microscopy. These findings suggest that the routine use of microscopy should be re-evaluated, particularly given its limited diagnostic benefit in cases where PCR or culture are negative.

This study has several limitations, including its retrospective design, the absence of standardized documentation of lesion characteristics, and its single-center setting, which may limit the generalizability of the findings. In addition, the stepwise approach to fungal detection, initial PCR testing followed by fungal culture for PCR-negative specimens, precludes an accurate comparison of the sensitivity and specificity between the two methods for dermatophyte detection.

In conclusion, this study provides a comprehensive overview of the epidemiology and trends of cutaneous fungal infections in Israel, emphasizing the predominance of *T. tonsurans* in upper-body skin sites among both adults and children. The rise in non-scalp *T. tonsurans* infections, particularly involving the face, neck, and upper trunk, reflects a shift in its clinical distribution. In countries where *T. tonsurans* is already a common cause of tinea capitis, continued surveillance is warranted to monitor its potential involvement in upper-body dermatophytoses. These findings support the need for ongoing epidemiologic monitoring and may inform clinical diagnostic and management strategies.

## Figures and Tables

**Figure 1 jof-11-00320-f001:**
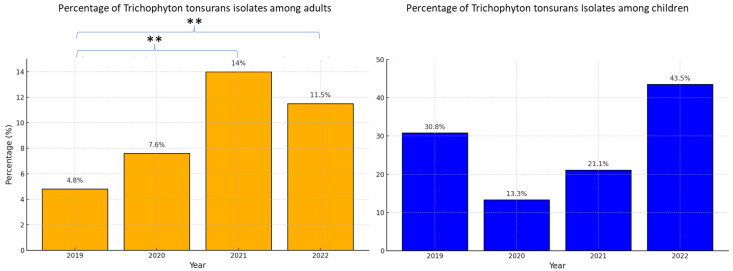
Yearly proportion of *Trichophyton tonsurans* among all confirmed fungal infections (2019–2022), stratified by age group (adults and children). ** *p*-value < 0.001.

**Figure 2 jof-11-00320-f002:**
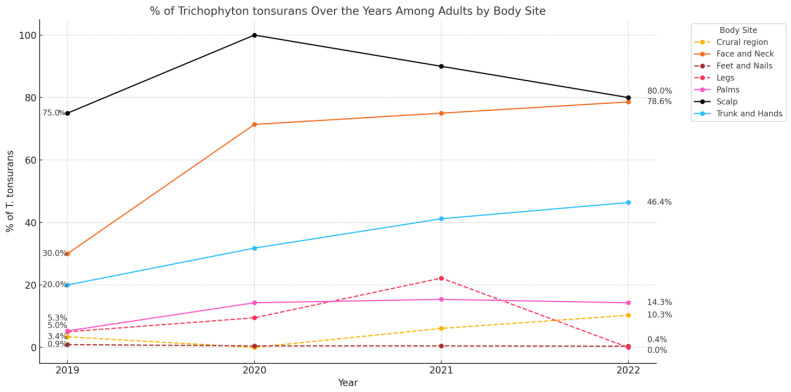
Anatomical distribution of *Trichophyton tonsurans* isolates in adults.

**Table 1 jof-11-00320-t001:** Demographic characteristics and positive detection rates of fungal infections by anatomical site.

	Nails	Feet	Scalp	Crural Region	Face and Neck	Palm	Trunk and Limbs
Age, mean (SD), years	46.6 (20.4)	46.5 (20.3)	19.8 (13.3)	49.8 (22.0)	32.4 (18.7)	47.6 (22.3)	48.3 (24.0)
Sex							
%Male	60.1%	63.2%	85.9%	71.9%	58.6%	79.2%	64.5%
%Female	39.9%	36.8%	14.1%	28.1%	41.4%	20.8%	35.5%
Positive detection rate ^a^, No. (%)	671 (61.4%)	397 (70.8%)	71 (39.4%)	114 (47.5%)	58 (38.7%)	48 (34.3%)	183 (33.8%)
% adults	97.2%	97.5%	67.7%	98.2%	87.9%	95.8%	96.2%
% children	2.8%	2.5%	32.4%	1.8%	12.1%	4.2%	3.8%

^a^ Positive detection rate is defined as the proportion of specimens confirmed by Polymerase Chain Reaction (PCR) and/or fungal culture out of all assessed specimens per anatomical site.

**Table 2 jof-11-00320-t002:** Distribution of fungal pathogens by body site in adults with confirmed cutaneous fungal infections.

	Overall (*n* = 1472)	Onyco-Mycosis (*n* = 652)	Feet (*n* = 387)	Crural Region (*n* = 112)	Scalp (*n* = 48)	Face and Neck (*n* = 51)	Palms (*n* = 46)	Trunk and Limbs (*n* = 176)
Dermatophytes, *n* (%)	1401 (95.2%)	607 (93.1%)	378 (97.7%)	105 (93.7%)	48 (100%)	51 (100%)	38 (82.6%)	175 (99.4%)
*Epidermophyton* *Fluccosum*	36 (2.4%)	16 (2.5%)	14 (3.6%)	2 (1.8%)	1 (2.1%)	1 (2.0%)	0	2 (1.1%)
*Microsporum* *Canis*	17 (1.2%)	5 (0.8%)	1 (0.3%)	3 (2.7%)	0	2 (3.9%)	1 (2.2%)	5 (2.8%)
*M. gypseum*	4 (0.3%)	1 (0.2%)	0	1 (0.9%)	0	0	1 (2.2%)	1 (0.6%)
*Trichophyton* *Rubrum*	1183 (80.4%)	570 (87.4%)	357 (92.2%)	90 (80.4%)	4 (8.3%)	13 (25.5%)	30 (65.2%)	119 (67.6%)
*T.* *mentagrophytes*/ *interdigitale*	20 (1.4%)	10 (1.5%)	3 (0.8%)	3 (2.7%)	0	1 (2.0%)	1 (2.2%)	2 (1.1%)
*T. tonsurans*	136 (9.2%)	4 (0.6%)	2 (0.5%)	6 (5.4%)	40 (83.3%)	35 (66.7%)	5 (10.9%)	45 (25.6%)
*T. violaceum*	5 (0.3%)	0	1 (0.3%)	0	3 (6.3%)	0	0	1 (0.6%)
*Candida*, *n* (%)	48 (3.3%)	30 (4.8%)	3 (0.8%)	7 (6.3%)	0	0	8 (17.4%)	0
*Albicans*	9 (0.6%)	1 (0.2%)	1 (0.3%)	6 (5.4%)	0	0	1 (2.2%)	0
*Parapsilosis*	36 (2.4%)	28 (4.3%)	2 (0.5%)	1 (0.9%)	0	0	5 (10.9%)	0
Other	3 (0.2%)	1 (0.2%)	0	0	0	0	2 (4.3%)	0
Non-dermatophyte molds, *n* (%)	23 (1.6%)	16 (2.5%)	6 (1.6%)	0	0	0	0	1 (0.6%)

**Table 3 jof-11-00320-t003:** Distribution of fungal pathogens by body site in children with confirmed cutaneous fungal infections.

	Overall (*n* = 70)	Onyco-Mycosis (*n* = 19)	Feet (*n* = 10)	Crural Region (*n* = 2)	Scalp (*n* = 23)	Face and Neck (*n* = 8)	Palms (*n* = 2)	Trunk and Limbs (*n* = 7)
Dermatophyte	69 (98.6%)	19 (100%)	10 (100%)	1 (50%)	23 (100%)	7 (100%)	2 (100%)	7 (100%)
*Epidermophyton* *Fluccosum*	1 (1.4%)	0	0	1 (50%)	0	0	0	0
*Microsporum* *Canis*	11 (15.7%)	0	0	0	9 (39.1%)	1(14.3%)	0	1 (14.3%)
*Trichophyton* *Rubrum*	36 (51.4%)	19 (100%)	10 (100%)	0	2 (8.7%)	2 (28.6%)	2 (100%)	1 (14.3%)
*T. tonsurans*	20 (28.6%)	0	0	0	11 (47.8%)	4 (57.1%)	0	5 (71.4%)
*T. violaceum*	1 (1.4%)	0	0	0	1 (4.3%)	0	0	0
*Candida*	1 (1.4%)	0	0	1 (50%)	0	0	0	0
*Albicans*	1 (1.4%)	0	0	1 (50%)	0	0	0	0

## Data Availability

The data supporting the findings of this study are available from the corresponding author upon request owing to privacy/ethical restrictions.

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
