# Peer review of "Epidemiology and Trends of Cutaneous Fungal Infections (2019–2022) in Israel: A Single Tertiary-Center Study"

_jof, 2025, doi:10.3390/jof11040320_

Round 1
Reviewer 1 Report
- More informative introduction is recommended to elaborate the current studies on Trichophyton tonsurans trends.
- The data describing in the results session about the table 2 does not match the data shown in the table 2.
- Multiple quotations are missing throughout the manuscript.
- While the conclusion holds itself, it would be more accurate to point out the decrease of 2022 since the article focusing on the trends of Trichophyton tonsurans.
- In line 32-33, the authors describe the fungal infection in immunocompetent individuals, it would be nice if the authors could include the information on immunocompromised population.
- In line 43-44, the authors briefly introduced the emerging of T. tonsurans infection, which is the fundamental information driving the research of this article. It would be more informative if the authors could elaborate on the when does it start? does the emergence of the infection cover the 2019-2022 time period? in what region?
- in line 108-109, the data described to be found in table 2 could not be found in the table 2.
- in line 120-125, multiple data described do not match the data in the table 2.
- in line 126, it's not the second and third since they have the same percentage.
- in line 127-128, in the text, n of Trichophyton tonsurans is 5 while it's 1 in the table 2.
- in figure 1, one is 'trichophyton tonsurans' and the other one is 'Trichophyton tonsurans'.
- in line 176-177, quotation needed.
- in line 185, 'scalp. reinforcing' is a grammar issue.
- in line 192, quotation needed.
- in line 217, for the abbreviations, fungal names should be added to the abbreviations.
Author Response
We thank the reviewer for his constructive comments, which have helped improve the clarity and scientific value of our manuscript. Below, we provide a detailed, point-by-point response to each comment.
Comment 1: In lines 32-33, the authors describe the fungal infection in immunocompetent individuals; it would be nice if the authors could include information on the immunocompromised population.
Response: We thank the reviewer for this suggestion. We have added a sentence to the introduction discussing the clinical course of fungal infections in immunocompromised patients, including those with HIV, malignancies, and primary immunodeficiencies such as CARD9 deficiency (lines 8–11).
Comment 2: In lines 43-44, the authors briefly introduce the emergence of T. tonsurans infection. It would be more informative if the authors could elaborate on when it started, whether the emergence covers the 2019–2022 period, and in what region.
Response: We have expanded the introduction and discussion to clarify that T. tonsurans has been increasingly reported since the early 2010s in pediatric patients with tinea capitis in Israel, with references to local and international trends. We also specify that its rise is captured within the 2019–2022 study period (lines 22–34).
Comment 3: In Figure 1, T. tonsurans infection in adults rises from 2019 to 2021 and decreases in 2022. The authors should discuss the possibility of a downward trend.
Response: This point has been addressed in the Results section (line 3.5) and discussed in the Discussion, where we note that the slight decrease in 2022 was not statistically significant and might represent normal annual variability.
Comment 4: In Figure 1, the 'T' in Trichophyton is inconsistent.
Response: We have corrected the inconsistency in capitalization across all figures and text.
Major Comments:
Comment 1: A more informative introduction is recommended to elaborate on current studies of T. tonsurans trends.
Response: We have revised the introduction and discussion to incorporate recent data and global comparisons on T. tonsurans, including local and global studies.
Comment 2: The data described in the Results section about Table 2 does not match the table.
Response: We reviewed and corrected inconsistencies. The text now matches the data in Table 2.
Comment 3: Multiple quotations are missing.
Response: We have added citations throughout the Introduction, Methods, Results, and Discussion sections to support our statements.
Comment 4: The conclusion should mention the 2022 decrease since the study focuses on trends.
Response: We have added this point explicitly in both the Results and Discussion sections.
Detailed Comments:
1. Line 108-109: The data described to be found in Table 2 could not be found.
Response: We corrected the text to align with the table.
2. Lines 120-125: Data described does not match Table 2.
Response: Fixed. Data now matches across both.
3. Line 126: It's not the second and third since they have the same percentage.
Response: Corrected for clarity.
4. Line 127-128: In the text, n for T. tonsurans is 5; it's 1 in Table 2.
Response: Corrected to match Table 2.
5. Figure 1: Capitalization inconsistent.
Response: Fixed.
6. Line 176-177: Quotation needed.
Response: Citation added.
7. Line 185: 'scalp. reinforcing' is grammatically incorrect.
Response: Sentence revised.
8. Line 192: Quotation needed.
Response: Citation added.
9. Line 217: Abbreviations should include fungal names.
Response: Added to the abbreviation list.
Reviewer 2 Report
The article is interesting, however, to be published it requires important changes.
In the abstract, authors should eliminate the use of excessive data (i.e. p-value, frequency changes, etc.)
In the title, authors should add the site where the study is conducted
Line 34 : Add a reference
In the introduction, authors should include a paragraph describing the genus that causes dermatophytes. They should also show the advantages and disadvantages of the methods used to identify dermatophytes. Finally a state of the art of the situation of cutaneous fungal infections in Israel.
The authors do not explain how the patients were selected for their study. Furthermore, they use a more sensitive technique such as PCR to discriminate the cases and subsequently evaluate the cases by less sensitive techniques such as microscopy and culture. This should be the other way around.
The authors do not explain how many samples were obtained from patients. Also, the authors do not describe how NDM and Candida are identified.
Authors must cite the reference that uses the PCR kit.
Authors should use the kappa index to determine the agreement between the methods used.
Table 1 does not describe the method by which positive cases were evaluated.
The results are difficult to interpret because they are not clearly explained. It is a lot of information for the reader that cannot be the logical order of the results.
The results section should be rewritten.
It would be very interesting if the authors described the characteristics of the infections and could compare them between adults and children. This could bring novelty to the work.
The discussion is poor and the authors should compare their results with those obtained by others. The paper does not describe the limitations of the study.
The work also does not discuss the usefulness of the methods for comparing the different species identified, much less their coincidence.
The titles of tables and graphs need to be improved.
Figure 2 shows only the results in adults, however in children it could be interesting to analyze them.
The data is interesting but the results must be presented and analyzed better.
The article is interesting, however, to be published it requires important changes.
In the abstract, authors should eliminate the use of excessive data (i.e. p-value, frequency changes, etc.)
In the title, authors should add the site where the study is conducted
Line 34 : Add a reference
In the introduction, authors should include a paragraph describing the genus that causes dermatophytes. They should also show the advantages and disadvantages of the methods used to identify dermatophytes. Finally a state of the art of the situation of cutaneous fungal infections in Israel.
The authors do not explain how the patients were selected for their study. Furthermore, they use a more sensitive technique such as PCR to discriminate the cases and subsequently evaluate the cases by less sensitive techniques such as microscopy and culture. This should be the other way around.
The authors do not explain how many samples were obtained from patients. Also, the authors do not describe how NDM and Candida are identified.
Authors must cite the reference that uses the PCR kit.
Authors should use the kappa index to determine the agreement between the methods used.
Table 1 does not describe the method by which positive cases were evaluated.
The results are difficult to interpret because they are not clearly explained. It is a lot of information for the reader that cannot be the logical order of the results.
The results section should be rewritten.
It would be very interesting if the authors described the characteristics of the infections and could compare them between adults and children. This could bring novelty to the work.
The discussion is poor and the authors should compare their results with those obtained by others. The paper does not describe the limitations of the study.
The work also does not discuss the usefulness of the methods for comparing the different species identified, much less their coincidence.
The titles of tables and graphs need to be improved.
Figure 2 shows only the results in adults, however in children it could be interesting to analyze them.
The data is interesting but the results must be presented and analyzed better.
Author Response
We thank the reviewer for his constructive comments, which have helped improve the clarity and scientific value of our manuscript. Below, we provide a detailed, point-by-point response to each comment. Reviewer comments are shown in bold, with our responses following each.
Comment 1: Authors should add the site where the study is conducted.
Response: The title has been revised to include the site of the study: “in Israel: A single tertiary-center study.”
Comment 2: Authors should include a paragraph describing the species that cause dermatophytes. They should also show the advantages and disadvantages of the methods used to identify dermatophytes (culture, microscopy and PCR). Finally, a state of the art of the situation of cutaneous fungal infections in Israel.
Response: We revised the Introduction to include:
- A paragraph describing the three main genera of dermatophytes: Trichophyton, Microsporum, and Epidermophyton.
- A discussion of the pros and cons of PCR, culture, and microscopy.
- An overview of recent trends in cutaneous fungal infections in Israel, with updated references added.
Comment 3: The authors do not explain how the patients were selected for their study.
Response: We clarified in the Methods that this is a retrospective study including all patients who presented with suspected cutaneous fungal infections at Sheba Medical Center between 2019 and 2022.
Comment 4: The authors use a more sensitive technique (PCR) first and then follow up with less sensitive techniques (microscopy and culture). This should be the other way around.
Response: As this is a non-interventional retrospective study based on routine clinical practice, PCR was performed first per institutional protocol. We added rationale for this approach, highlighting that culture was used in PCR-negative cases to improve species identification.
Comment 5: The authors do not explain how many samples were obtained from patients.
Response: We specified the total number of PCR tests (2800), fungal cultures (905), and microscopy tests (897) performed (Results section 3.1).
Comment 6: The authors do not describe how NDM and Candida are identified.
Response: We added to the Methods that Candida and non-dermatophyte molds (NDMs) were identified by fungal culture.
Comment 7: Authors must cite the reference that uses the PCR kit.
Response: We cited the relevant reference describing the DERMADYN multiplex PCR kit (Reference 24).
Comment 8: Authors should use the kappa index to determine agreement between the methods used.
Response: We calculated and reported the kappa index (κ = 0.41) to describe the level of agreement between PCR and fungal culture (Results section 3.3).
Comment 9: Table 1 does not describe the method by which positive cases were evaluated.
Response: We added a footnote to Table 1 clarifying that “positive detection” refers to confirmation by PCR and/or fungal culture.
Comment 10: The results are difficult to interpret because they are not clearly explained. The section should be rewritten.
Response: We restructured the Results section into logical subsections, clarified the flow of information, and ensured that data in the text and tables are consistent.
Comment 11: Describe the characteristics of the infections and compare between adults and children.
Response: Clinical features such as lesion size, shape, and number were not consistently documented and could not be retrieved retrospectively. Furthermore, the small pediatric sample limited statistical power. These limitations are now acknowledged in the Discussion.
Comment 12: The discussion is poor and lacks comparison to other literature.
Response: We expanded the Discussion to include comparative studies highlighting similarities and differences in trends, species prevalence, and anatomical distribution.
Comment 13: The paper does not describe the limitations of the study.
Response: We added a paragraph to the Discussion explicitly describing study limitations.
Comment 14: The work does not discuss the usefulness of the methods for comparing the different species identified.
Response: We now discuss the performance of PCR vs. fungal culture, their concordance, and diagnostic utility. The moderate agreement (κ = 0.41) and added value of culture for PCR-negative cases are highlighted.
Comment 15: The titles of tables and graphs need to be improved.
Response: We revised and standardized all figure and table titles and legends for clarity and consistency.
Comment 16: Figure 2 shows only the results in adults. However, in children it could be interesting to analyze them.
Response: Due to the limited number of pediatric cases per anatomical site, generating a meaningful figure was not feasible. However, we included a supplemental figure for the reviewers’ reference in case they consider it valuable to add to the main manuscript.
Comment 17: The English could be improved to more clearly express the research.
Response: The manuscript underwent professional English language editing by a certified editing service, and the certificate has been provided. Additionally, a further round of English proofreading was conducted for the current revised version to ensure clarity and fluency throughout the text.
Comment 18 (Abstract): Authors should eliminate excessive data (p-values, frequency changes, etc.)
Response: Done.
We hope these revisions meet the expectations of the reviewers and editors. We appreciate the opportunity to improve our manuscript and look forward to your further evaluation.
Sincerely,
Dr. Eran Galili
(On behalf of all authors)
Round 2
Reviewer 1 Report
The study highlights the increased emergence of T. tonsurans in Israel from 2019 to 2022 in cutaneous fungal infections, especially in upper body skin site. Thank you for the one by one response and corrections.
N/A
Author Response
Comment 1: The study highlights the increased emergence of T. tonsurans in Israel from 2019 to 2022 in cutaneous fungal infections, especially in upper body skin site. Thank you for the one by one response and corrections.
Response: We sincerely thank the reviewer for the positive feedback and for acknowledging the relevance of our findings. We appreciate your careful review and are pleased that our point-by-point responses and revisions have addressed your concerns.
Reviewer 2 Report
The article improved greatly after the initial review. However, other details need to be addressed by the authors.
Authors only delete the numbers on line 19, as well as the p-value.
In the materials and methods section, there are still concepts to be clarified, such as: Confirmed mycosis. Furthermore, it would be interesting to compare microscopy with culture and PCR. The authors do not specify the values for the Kappa index, so they could say that the concordance is moderate, and they do not discuss what this value means, nor its impact.
The sentence that begins on line 124 and ends on line 127 is very long. The idea is not understood.
There are still microorganism names that are not written in italics, e.g. Line 198
In the discussion there are some ideas that should be added: explain the changes in children of those described for T. tonsurans.
The differences between microscopy and PCR are not explained. This is an important element of the text.
Authors should conclude the article with the impact of the work and what their study means.
The article improved greatly after the initial review. However, other details need to be addressed by the authors.
Authors only delete the numbers on line 19, as well as the p-value.
In the materials and methods section, there are still concepts to be clarified, such as: Confirmed mycosis. Furthermore, it would be interesting to compare microscopy with culture and PCR. The authors do not specify the values for the Kappa index, so they could say that the concordance is moderate, and they do not discuss what this value means, nor its impact.
The sentence that begins on line 124 and ends on line 127 is very long. The idea is not understood.
There are still microorganism names that are not written in italics, e.g. Line 198.
In the discussion there are some ideas that should be added: explain the changes in children of those described for T. tonsurans.
The differences between microscopy and PCR are not explained. This is an important element of the text.
Authors should conclude the article with the impact of the work and what their study means.
Author Response
We sincerely thank the reviewer for acknowledging the improvements made following the initial review. We appreciate your continued engagement and careful evaluation of our work. In the current revision, we have thoroughly addressed all remaining points raised and made further refinements to enhance the clarity, precision, and scientific value of the manuscript. We hope these updates meet your expectations.
Comment 1:
“In the Materials and Methods section, there are still concepts to be clarified, such as: Confirmed mycosis.”
Response:
We agree with this important point. In the Materials and Methods section, we have now explicitly defined “confirmed cutaneous mycosis” as “identification of a fungal pathogen via PCR and/or fungal culture.”
Comment 2:
“Furthermore, it would be interesting to compare microscopy with culture and PCR.”
Response:
We appreciate this suggestion and have added a comparative analysis between all three diagnostic methods. Specifically, we included:
- PCR vs. culture (κ = 0.41)
- PCR vs. microscopy (κ = 0.54)
- Microscopy vs. culture (κ = 0.48)
All indicate moderate agreement, and the implications of these findings are discussed in both the Results (Section 3.3) and Discussion sections.
Comment 3:
“The authors do not specify the values for the Kappa index, so they could say that the concordance is moderate, and they do not discuss what this value means, nor its impact.”
Response:
We have now specified the Cohen’s kappa index values and included a standard interpretation scale in the Statistical Analysis subsection. Additionally, we discuss the clinical relevance of the moderate agreement in the Discussion, emphasizing that while PCR and culture are complementary, they are not interchangeable.
Comment 4:
“The sentence that begins on line 124 and ends on line 127 is very long. The idea is not understood. The differences between microscopy and PCR are not explained. This is an important element of the text.”
Response:
Thank you for this important observation. We have revised the sentence to improve clarity and now clearly explain the rationale and diagnostic differences between PCR and microscopy. We also conclude in the Discussion that, based on our findings, the diagnostic benefit of microscopy in cases already evaluated by PCR or culture is limited.
Comment 5:
“Authors should conclude the article with the impact of the work and what their study means.”
Response:
We have added a concluding paragraph to the Discussion section. It summarizes the clinical and epidemiological impact of our findings, with emphasis on the rise in non-scalp T. tonsurans infections. We also highlight the importance of continued surveillance and improved diagnostic awareness—especially in settings where T. tonsurans tinea capitis is already common.
Comment 6:
“Authors only delete the numbers on line 19, as well as the p-value.”
Response:
This has been corrected. The abstract no longer includes specific percentages or p-values, in accordance with your recommendation.
Comment 7:
“There are still microorganism names that are not written in italics, e.g. Line 198.”
Response:
We have thoroughly reviewed the entire manuscript and corrected all microorganism names to appear in italics, including the one referenced.
Comment 8:
“In the discussion there are some ideas that should be added: explain the changes in children of those described for T. tonsurans.”
Response:
We appreciate this suggestion. In the revised Discussion, we now explicitly address the broader anatomical distribution of T. tonsurans among children. We note that it was frequently isolated from non-scalp sites such as the face, neck, and trunk, and we include relevant references to contextualize this finding.